# Risk-adjusted trend in national inpatient fall rates observed from 2011 to 2019 in acute care hospitals in Switzerland: a repeated multicentre cross-sectional study

Niklaus S Bernet [1], Irma H J Everink,[2] Sabine Hahn,[1] Marianne Müller,[1] Jos M G A Schols[2]

¹School of Health Professions, Bern University of Applied Sciences, Bern, Switzerland
²Department of Health Services Research, Maastricht University; Care and Public Health Research Institute, Maastricht, The Netherlands

**Correspondence to**
Niklaus S Bernet;
niklausstefan.bernet@bfh.ch

## ABSTRACT

**Objectives** This study aimed to investigate whether a significant trend regarding inpatient falls in Swiss acute care hospitals between 2011 and 2019 could be confirmed on a national level, and whether the trend persists after risk adjustment for patient-related fall risk factors.

**Design** A secondary data analysis was conducted based on annual multicentre cross-sectional studies carried out between 2011 and 2019.

**Setting** All Swiss acute care hospitals were obliged to participate in the surveys. Except for emergency departments, outpatient wards and recovery rooms, all wards were included.

**Participants** All inpatients aged 18 or older who had given their informed consent and whose data were complete and available were included.

**Outcome measure** Whether a patient had fallen in the hospital was retrospectively determined on the survey day by asking patients the following question: Have you fallen in this institution in the last 30 days?

**Results** Based on data from 110 892 patients from 222 Swiss hospitals, a national inpatient fall rate of 3.7% was determined over the 9 survey years. A significant linear decreasing trend (p=0.004) was observed using the Cochran-Armitage trend test. After adjusting for patient-related fall risk factors in a two-level random intercept logistic regression model, a significant non-linear decreasing trend was found at the national level.

**Conclusions** A significant decrease in fall rates in Swiss hospitals, indicating an improvement in the quality of care provided, could be confirmed both descriptively and after risk adjustment. However, the non-linear trend, that is, an initial decrease in inpatient falls that flattens out over time, also indicates a possible future increase in fall rates. Monitoring of falls in hospitals should be maintained at the national level. Risk adjustment accounts for the observed increase in patient-related fall risk factors in hospitals, thus promoting a fairer comparison of the quality of care provided over time.

## STRENGTHS AND LIMITATIONS OF THIS STUDY

⇒ The internationally standardised primary data collection used in this study ensures high data quality in terms of completeness and uniformity, offering relevant advantages compared with the use of data generated for other purposes (eg, a critical incidence reporting system).

⇒ A comprehensive, nationally representative sample was used, comprising almost all Swiss hospitals with a somatic acute care mandate.

⇒ The risk adjustment method used enables a fairer comparison of inpatient fall rates over time, as it controls for changes in patient-related fall risk factors.

⇒ The cross-sectional survey design hinders consideration of the temporal sequence between inpatient fall risk factors and fall events, thereby limiting the recommended exclusive focus on pre-existing fall risk factors that cannot be influenced by hospital care in the risk adjustment model.

measuring and reporting quality indicators at a national level to inform stakeholders at different levels of the health system—is assumed to drive quality improvement processes. This happens via two main pathways, as outlined by Berwick *et al*.[1] First, the 'improvement by choice' pathway allows healthcare consumers to select providers based on publicly reported performance data. This incentivises providers to improve the quality of care due to market competition.[1 2] Second, the 'improvement through change' pathway uses performance data to highlight areas where quality improvements are needed, fostering intrinsic motivation for providers to initiate improvements.[1 2] In addition, the fear of reputational damage from making performance data publicly available

## INTRODUCTION

Regular national quality measurement in healthcare—the process of systematically

can be a further incentive to improve the quality of healthcare services.[3 4]

National quality measurement is, therefore, a recognised strategy for monitoring and improving the quality of healthcare by national policy-makers.[3 5] Regular quality measurement provides the basis for initiating data-driven quality improvement strategies, such as benchmarking and public reporting or the monitoring of achievements over time.[3 5] There is also some indication in the literature that these strategies can have a positive effect on the quality of care over time.[6 7] For example, a risk-adjusted trend analysis in the USA between 2010 and 2019 showed a significant reduction in adverse events, particularly in hospitals that were affected by targeted quality improvement efforts during this period.[8]

One of the quality indicators that could be improved through regular national measurements and targeted quality improvement efforts is the inpatient fall rate. Falls in the hospital have serious consequences for both patients and the healthcare system as a whole.[9 10] Based on data from Australia, a fall in hospital results, on average, in an 8-day longer hospital stay and an additional cost, in purchasing power parity adjusted international dollars,[11] of Int\$4864.[12] Many inpatient falls, though not all, can be prevented with the rigorous application of best-practice prevention measures.[13] Therefore, the number of falls in hospital is widely recognised as a relevant indicator of quality of care. Internationally, there are only a few organisations that we are aware of, such as the Press Ganey National Database of Nursing Quality Indicators (NDNQI)[14 15] or the International Prevalence Measurement of Care Quality (LPZ),[16 17] that provide a national infrastructure for collecting data on the quality indicator of falls in hospital.

Since 2011, the LPZ measurement has been conducted annually in Switzerland as a national mandatory quality measurement initiative, commissioned by the Swiss Association for Quality Development in Hospitals and Clinics (ANQ). Alongside other indicators, the prevalence of inpatient falls in all Swiss acute care hospitals is determined using cross-sectional primary data.[18] The aim is multifaceted: to inform a wide range of stakeholders—politicians, hospitals and the general public—about the findings and, in particular, to provide hospitals with the data needed to initiate quality improvement measures.[19] Data collected per measurement are forwarded to the Bern University of Applied Sciences (BFH) for comprehensive national analysis and reporting. The results will ultimately be published on the ANQ website (www.anq.ch), both in the form of a detailed report and a transparent national hospital comparison.[18–20] A recently submitted descriptive analysis based on these data from 2011 to 2019 showed that inpatient fall rates in Switzerland decreased substantially after the first survey and then stagnated over time.[21] However, this putative trend could not be conclusively assessed purely on the basis of the descriptive analysis. A decreasing trend in fall rates at the national level might be an indication that the quality of care in hospitals with regard to fall prevention has improved over time. However, a stagnation of fall rates over time would contradict the hypothesis outlined by Berwick et al[1] that regular quality measurements favour the initiation of quality improvement processes and thus continuous quality improvement, which would be reflected in decreasing national inpatient fall rates over time. It is, however, also possible that there are other reasons that favour a stagnation of fall rates in the hospitals and thus possibly mask the effect of regular quality measurements on quality improvement in the hospitals.

► Not all falls are preventable: Even if fall prevention is continuously improved, the fall rates will remain constant at a certain level, as a reduction to zero will not be achievable.[13]
► Lack of resources to intensify fall prevention: Due to increasing financial constraints and staff shortages, fall prevention cannot be further strengthened.[22]
► Patient-related risk factors for falls are increasing: More and more often, older and multimorbid patients are being treated in the hospital, that is, patients who carry a higher risk of falling. This means that more patients with a higher fall risk profile are being treated in the hospital over time. As a result, falls in the hospital do not decrease despite constant quality improvements.[21]

In the last scenario, keeping fall rates constant would still be a success.

In order to accurately compare inpatient fall rates over time while accounting for changing patient-related fall risk factors, it is usually recommended to conduct a risk-adjusted comparison.[23] Risk adjustment in this context means that a statistical model is used to control for ongoing changes in patient-related fall risk factors that cannot be influenced by hospitals. If patient-related fall risk factors are kept constant over time, it can be ensured that positive/negative trends are actually due to quality improvements/deteriorations in care rather than to changes in patient-related risk factors.

The objectives of this study were as follows:
► First, to investigate whether a trend regarding the prevalence of inpatient falls in acute care hospitals between 2011 and 2019 can be statistically confirmed on a national level in Switzerland.
► Second, whether an identified trend persists after risk adjustment for patient-related fall risk factors.

## METHODS
### Study design
We conducted a secondary data analysis based on multicentre cross-sectional primary data collected annually between 2011 and 2019 as part of the LPZ measurement in Switzerland. This measurement is used to collect data on various quality indicators, such as falls, pressure injuries and malnutrition, in different countries (eg, the Netherlands, Austria and Switzerland) during a defined data collection period. In Switzerland, usually on the Tuesday of the second week in November. The methods

used in the so-called LPZ measurement are described in detail elsewhere.[17 18 24]

## Data collection and population

The LPZ questionnaire was used to collect the primary data.[17 18] This questionnaire emerged from an instrument developed on the basis of a literature review, a Delphi and a pilot study for the cross-setting and cross-institutional survey of pressure injuries,[25] which was assessed as reliable and valid.[25–27] Subsequently, other instruments were added to the initial questionnaire to capture further quality indicators, such as falls, by an international research group and based on evidence-based research. The LPZ questionnaire is divided into three parts: institutional, ward and patient questionnaires. In 2016, it underwent a revision and was subsequently referred to as LPZ 2.0. This revision streamlined the questionnaire by reducing the number of structure and process indicators while largely preserving the questions related to patient characteristics. Although the revision also affected the questions on quality indicators, these changes involved a reduction in the number of questions rather than a change in content. For instance, regarding the quality indicator on falls, specific context questions on the time and place of the fall and the main causes were deleted. Detailed descriptions of the adjustments made to the 'inpatient fall' variable can be found in the paragraph 'outcome variable'.

Data collection in the hospitals was organised and coordinated by designated hospital coordinators, who were trained during annual national training events using the train-the-trainer principle, so that they could subsequently train their clinical data collection teams on site. To additionally ensure uniform data collection, a detailed measurement manual with definitions, instructions, explanations and examples was available to all persons involved. The clinical data collection teams each consisted of two nurses. They gathered the required data by directly questioning the patient or, where permissible, from the patient's documentation on the survey day and then entered it into the web-based data entry programme.

In Switzerland, all acute care hospitals with an acute somatic service mandate have been obliged to collect data on at least pressure injuries and inpatient falls once a year in November since 2011. Other institutions were allowed to participate in the LPZ measurement on a voluntary basis. In 2020 and 2021, no LPZ measurement took place in Switzerland due to the COVID-19 pandemic. At the ward level, all wards were included, with the exception of emergency departments, outpatient wards and recovery rooms as of 2012, and additionally, maternity wards as of 2013. On the survey day, all inpatients were included in the LPZ measurement. In addition, written consent had to be obtained from patients or their legal representatives for inclusion in 2011, and verbal consent from 2012 onwards (see the ethics statements). The hospital coordinators were responsible for ensuring that all patients or, if they were incapacitated, their legal representatives

were informed in advance, in writing, about the aim and procedure of the measurement. The written/oral consent of the patients was clarified directly at the time of the survey, and that of the legal representatives in advance.

For the national analysis and reporting, the BFH adjusted the raw LPZ dataset for each measurement to fulfil the inclusion and exclusion criteria specified by ANQ for national reporting.

The following adjustments were made:
▶ Exclusion of institutions that do not have an acute somatic service mandate, such as psychiatric, rehabilitation and geriatric clinics.
▶ Exclusion of patients under the age of 18.

## Study population

For the present study, we had full access to the national analysis datasets of the BFH from 2011 to 2019, to which the following general adjustments were made for the current secondary data analysis:
▶ Exclusion of conflicting cases.
▶ Exclusion of patients for whom it was unknown whether they had fallen in the institution in the last 30 days.
▶ Exclusion of non-participating patients from whom only the reason for non-participation was recorded.
▶ Exclusion of cases due to missing information on relevant patient-related fall risk variables.

See also online supplemental figure S1 for a flow chart depicting the selection of cases for analysis.

## Outcome variable

The outcome variable 'inpatient fall' is defined in the LPZ measurement as 'any unintentional change in position that results in the client coming to rest on the ground or other lower level, regardless of the reason'.[17 28]

To determine whether or not a fall occurred in the hospital, the assessment was carried out retrospectively by the clinical data collection teams on the day of the survey. The assessment process involved three steps: first, the patients were asked; second, the nurse in charge was consulted; and third, patient documentation was reviewed for any recorded falls.

The following question(s) with specified response categories had to be answered:
▶ From 2011 to 2015:
'How often has the patient fallen in the last 30 days?' with responses categorised as once (1), twice (2), thrice (3), more than three times (4), unknown (5) and not fallen (6).
'When was the most recent fall?' with responses categorised as before admission in current healthcare facility (1) or after admission (2).
▶ From 2016 to 2019, after revision of the LPZ questionnaire, the questions were simplified to:
'Has the patient fallen in the last 30 days in this institution?' with responses being no (0), yes (1) and unknown (97).

 

Based on the raw variables, the outcome variable 'inpatient fall' was calculated as follows for the present study:

► From 2011 to 2015:

The variable on the frequency of falls in the last 30 days was recoded as follows: 6 (no fall) to 0; 1–4 to 0 if the variable on timing of the last fall equals 1 (before admission); 1–4 to 1 if variable for timing of the last fall equals 2 (after admission). In this way, a new variable was created indicating whether the patient had fallen (yes (1) or no (0)) in the last 30 days in this institution. In addition, due to the lack of information, all cases with 'unknown' (5) concerning the question of the frequency of falls in the last 30 days were excluded from the study sample.

► From 2016 to 2019:

The new variable indicating whether the client had fallen (no or yes) in the last 30 days in this institution was created by recoding the original variable no (0) to 0, yes (1) to 1 and unknown (97) to missing. In addition, all cases with missing values on the newly created variable were excluded from the study sample due to missing information on whether the patient had fallen in the last 30 days in this institution.

### Patient-related fall risk variables (covariates)

The LPZ questionnaires used between 2011 and 2019 were studied to identify potential patient-related characteristics that could be used for risk adjustment. To be considered as potential variables, they had to fulfil the following criteria:

► The variable was collected on all survey dates.
► The variable had not undergone any significant changes in content over the course of the survey years (in some cases, minor linguistic changes were accepted).

The following patient-related characteristics were identified as possible variables for risk adjustment: age, sex, surgical procedure within 14 days prior to measurement day, care dependency according to the Care Dependency Scale (CDS), ICD-10 (International Statistical Classification of Diseases and Related Health Problems 10th Revision[29]) diagnosis groups. For more information on the variables used in this study, see table 1.

### Data analysis

The datasets from each of the nine survey years were prepared by standardising the variable names, descriptions and labels. Following this, case exclusion criteria were applied consistently, as detailed in the flow chart in online supplemental figure S1, and the outcome variable was calculated uniformly across all datasets. After merging the datasets into one overall dataset by adding a time variable, the sample was first described by means of frequencies and percentages, and age by means of median and IQR due to the skewed distribution. Furthermore, the national prevalence of falls in the hospital between 2011 and 2019 was estimated as described by Thomann *et al*.[20] For this purpose, using the dataset for the corresponding year, the number of patients with a fall in the hospital was divided by the total number of patients in the dataset and then multiplied by 100 to obtain the inpatient fall rate as a relative frequency in per cent.

Second, to investigate whether a trend in the prevalence of inpatient falls in acute care hospitals could be statistically confirmed at the national level in Switzerland between 2011 and 2019, we started the analysis with the simple and still frequently used Cochrane-Armitage trend test. This test is suitable for identifying a linear trend or association between an ordinal dependent variable (time treated as an ordinal variable here) and a binary dependent variable, where the null hypothesis is that there is no trend,[30] that is, that fall rates have remained the same over time. For purposes of visualisation, a linear trend line was plotted based on a linear regression.

Third, to determine whether an identified trend persists after risk adjustment for patient-related fall risk factors, we performed steps a–f, with a–d covering risk adjustment model development and e–f describing reporting and visualisation:

1. As a starting point for setting up the risk adjustment model, we performed a logistic regression resulting in model 'A', which included all available patient-related variables as listed in table 1 as covariates and, to depict a trend, the time variable as a numerical covariate (coded from 1 to 9, where 1 denotes the survey year 2011 and 9 the survey year 2019). As it is rather unlikely that an increase or decrease will remain constant over the years, a quadratic time-effect was also included in model 'A' in order to be able to take account of a non-linear relationship. Additionally, we tested for interaction effects between patient-related fall risk variables and the time variable.

2. A stepwise backward variable selection algorithm based on the Akaike information criterion (AIC)[31] was then applied to model 'A' to determine the relevant patient-related fall risk factors to be included in the risk adjustment model and to check for the presence of a linear or a non-linear trend. The resulting model was designated as model 'B'.

3. Since the data have a hierarchical structure (patients grouped in the hospitals), the selected variables according to model 'B' were modelled as fixed effects and the hospitals as a random effect in a two-level random intercept logistic regression model (model 'C'). We wanted to make inferences regarding the model parameters of a random-effects model, not any finite population characteristics. This model-based approach, in contrast to a design-based approach, allowed us to generalise to a hypothetical population beyond the data set under consideration.[32]

4. To determine whether the model complexity of model 'C' could be further reduced, another two-level random intercept logistic regression model (model 'D') was calculated. In model 'D', the hospitals were modelled as a random effect and the statistically significant time-related factors as well as the patient-related fall

**Table 1** Overview of the variables used in the study

| Outcome variable | Answer options |
|---|---|
| Has the client fallen in the last 30 days in this institution | No (0)/yes (1) |

| Patient-related variables | Answer options |
|---|---|
| Age (in years) | Scale |
| Sex | Male (0)/female (1) |
| Surgical procedure within 14 days prior to measurement | No (0)/yes (1) |
| Care Dependency Scale (CDS)* | Care independent (70–75) (0); to a great extent independent (60–69) (1); partially dependent (45–59) (2); to a great extent dependent (25–44) (3); completely dependent (15–24) (4) |
| ICD-10—Diseases of the circulatory system | No (0)/yes (1) |
| ICD-10—Diseases of the musculoskeletal system and connective tissue | No (0)/yes(1) |
| ICD-10—Endocrine, nutritional and metabolic diseases | No (0)/yes (1) |
| ICD-10—Diseases of the genitourinary system | No (0)/yes (1) |
| ICD-10—Diseases of the digestive system | No (0)/yes (1) |
| ICD-10—Diseases of the respiratory system | No (0)/yes (1) |
| ICD-10—Neoplasms | No (0)/yes (1) |
| ICD-10—Mental, behavioural and neurodevelopmental disorders | No (0)/yes (1) |
| ICD-10—Diseases of the blood and blood-forming organs | No (0)/yes (1) |
| ICD-10—Diseases of the nervous system | No (0)/yes (1) |
| ICD-10—Certain infectious and parasitic diseases | No (0)/yes (1) |
| ICD-10—Diseases of the skin and subcutaneous tissue | No (0)/yes (1) |
| ICD-10—Injury, poisoning and certain other consequences of external causes | No (0)/yes (1) |
| ICD-10—Diseases of the eye and adnexa and ICD-10—Diseases of the ear and mastoid process† | No (0)/yes (1) |
| ICD-10—Congenital malformations, deformations and chromosomal abnormalities | No (0)/yes (1) |

| Time variable | Values |
|---|---|
| Time (survey year) | 2011 (1)/2012 (2)/2013 (3)/2014 (4)/2015 (5)/2016 (6)/2017 (7)/2018 (8)/2019 (9) |

*The CDS is composed of 15 questions that are rated on a scale from 1 to 5. The total score ranges from 15 to 75 points, with a lower score indicating higher care dependency.[65 66] The following five categories can be derived from the sum score: completely dependent on care from others, to a great extent dependent, partially dependent, to a great extent independent and care independent.[67 68]
†These ICD-10 diagnosis groups were only recorded separately from 2013 onwards and were, therefore, combined for the analyses in order to standardise them.
CDS, Care Dependency Scale; ICD-10, International Statistical Classification of Diseases, 10th Revision.

risk factors of model 'C' were included as fixed effects. If the analysis of variance test revealed no significant difference in the model fit between model 'C' and the reduced model 'D', model 'D' served as the final risk adjustment model that informed the subsequent steps e–f.

5. To determine whether the patient-related fall risk factors changed over time and therefore to get an idea of whether risk adjustment was necessary at all, the patient-related fall risk factors selected in model 'D'

(without time-related factors) were included as covariates in a two-level random intercept logistic regression model (model 'E'). The need for risk adjustment is indicated if the patient-related fall risk factors have increased or decreased over time. Such an increase/decrease can be confirmed if the national inpatient fall rates increase/decrease over time when predicted solely on the basis of the selected patient-related fall risk factors. The risk-adjusted national inpatient fall rates

predicted based on model 'E' were graphically plotted over time for visualisation purposes.

6. Finally, to demonstrate the predicted risk-adjusted trend in national fall rates in hospitals over time, the predicted risk-adjusted trend was graphically plotted on the basis of model 'D'. By controlling for the average effect of patient-related fall risk factors over time, changes in these factors can be largely excluded as a reason for a significant increasing/decreasing trend in the predicted national fall rates. Therefore, an increase/decrease, if any, would more likely be due to a deterioration/improvement in the quality of care provided in hospitals over time.

Fourth, an additional analysis was conducted by repeating the analyses on a subsample of patients aged 65 years and older. This subsample was selected because older patients in particular are more often subject to falls in the hospital.

Fifth, as part of a sensitivity analysis, we repeated the complete data analysis using only hospitals that provided data in all nine survey years to assess for potential bias in the results from hospitals that had joined or left the measurement during the study period. This approach was based on the assumption that the introduction of annual national quality measurement could trigger quality improvement initiatives in participating hospitals, potentially reducing fall rates over time. However, this trend could be masked by newer hospitals participating in the measurement, as they may initially have higher fall rates compared with hospitals already participating. The aim of the sensitivity analysis was, therefore, to detect such a masking effect if one existed.

Data cleaning and merging of the datasets were carried out with IBM SPSS Statistics (V.28). All analyses and graphics were carried out with the statistics programme R, V.4.1.0[33] and the packages 'DescTools',[34] 'MASS',[35] 'lme4',[36] 'effects'[37 38] and 'sjplot'.[39] In the analyses, a p<0.05 was set as the statistical significance level.

### Patient and public involvement
None.

## RESULTS
### Sample description
A total of 155 782 inpatients were recorded in 224 Swiss acute care hospitals on the survey days over the 9 survey years. Of these, 110 892 patients were included in the present study sample, yielding an average case inclusion rate of 71.2%. The case inclusion rate was lowest in 2011, with 36.1%, and highest in 2016, with 76.4%. The primary reasons for case exclusions were patient non-participation due to refusal, unavailability at the time of survey or cognitive impairment. Additionally, missing information on care dependency led to the exclusion of 4726 patients in 2011. Therefore, the results presented below are based on a total of 110 892 cases from 222

hospitals. Patient characteristics of the whole sample are described in table 2 and by survey year in online supplemental table S1.

## UNADJUSTED TREND IN NATIONAL INPATIENT FALL RATES
The national inpatient fall rate over the 9 years of the survey was 3.7%. As figure 1 shows, the national inpatient fall rate was highest in 2011, at 4.6%, and lowest in 2015, at 3.0%. At the last survey in 2019, the fall rate was 3.7%. Using the Cochran-Armitage trend test, a significant linear decreasing trend over time was found (p for trend=0.004). This means that the national fall rate decreased linearly over time when patient-related fall risk factors were not adjusted for. The estimated linear trend line (see also figure 1) intersects the intercept y at a predicted value of 4.15% in 2011 and reaches a predicted value of 3.41% in 2019.

The national inpatient fall rate in 2015 deviates from the rates of the other survey years. In a sensitivity analysis, we excluded the data from year 2015 and checked whether the results of the Cochran-Armitage trend test would become significantly different or not. Since a significant linear decreasing trend could be found both with and without the data from 2015, these were retained in the following analyses.

### Risk-adjusted trend of Swiss national inpatient fall rate
When patient-related fall risk factors were controlled for, a statistically significant non-linear trend, indicated by the negative coefficient (coeff.) of the time variable (–0.10, SE 0.03, p=0.001) and the positive coefficient of the squared time variable (0.01, SE 0.00, p=0.044) in the model, was observed in national inpatient fall rates (table 3). The model indicates that the relationship between time and inpatient falls at the national level is not linear, as the decreasing trend flattens out over time. The calculated ORs (online supplemental table S2) illustrate the observed non-linear trend in that, for example, at the national level, the odds of falling in the hospital decreased by 8% in 2012 compared with 2011 (OR 0.92, 95% CI 0.88 to 0.96). In contrast, however, the odds neither decreased nor increased in 2019 relative to 2018 (OR 1.00, 95% CI 0.96 to 1.04).

The trend described above was observed while controlling for a total of ten patient-related fall risk variables in the risk adjustment model (table 3): eight significantly fall risk-increasing patient-related variables, such as age (OR 1.02, 95% CI 1.01 to 1.02, p<0.001), being heavily care dependent (OR 5.69, 95% CI 5.10 to 6.36, p<0.001) or in the ICD-10 diagnosis group mental and behavioural disorders (OR 1.79, 95% CI 1.66 to 1.92, p<0.001); two significantly fall risk-reducing patient-related characteristics, sex (OR 0.82, 95% CI 0.77 to 0.87, p<0.001) and surgery in the last 14 days prior to measurement (OR 0.61, 95% CI 0.57 to 0.66, p<0.001).

Figure 2 illustrates the predicted national fall rates per survey year when the following are adjusted for: (a) only

**Table 2** Description of the patient characteristics of the sample

| Patient characteristics | n=110 892 | % |
|---|---|---|
| Sex (female) | 56 014 | 50.5 |
| Care dependency (CDS) | | |
| Care independent (70–75) | 59 874 | 54.0 |
| To a great extent independent (60–69) | 25 970 | 23.4 |
| Partially dependent (45–59) | 15 508 | 14.0 |
| To a great extent dependent (25–44) | 6992 | 6.3 |
| Completely dependent (15–24) | 2548 | 2.3 |
| Surgical procedure within 14 days prior to measurement (yes) | 47 233 | 42.6 |
| ICD-10—Diseases of the circulatory system (yes) | 58 986 | 53.2 |
| ICD-10—Diseases of the musculoskeletal system and connective tissue (yes) | 42 349 | 38.2 |
| ICD-10—Endocrine, nutritional and metabolic diseases (yes) | 34 244 | 30.9 |
| ICD-10—Diseases of the genitourinary system (yes) | 32 079 | 28.9 |
| ICD-10—Diseases of the digestive system (yes) | 27 884 | 25.1 |
| ICD-10—Diseases of the respiratory system (yes) | 25 600 | 23.1 |
| ICD-10—Neoplasms (yes) | 21 167 | 19.1 |
| ICD-10—Mental, behavioural and neurodevelopmental disorders (yes) | 19 827 | 17.9 |
| ICD-10—Diseases of the blood and blood-forming organs (yes) | 15 461 | 13.9 |
| ICD-10—Diseases of the nervous system (yes) | 13 494 | 12.2 |
| ICD-10—Certain infectious and parasitic diseases (yes) | 13 494 | 12.2 |
| ICD-10—Diseases of the eye and adnexa or ICD-10—Diseases of the ear and mastoid process (yes) | 8148 | 7.3 |
| ICD-10—Diseases of the skin and subcutaneous tissue (yes) | 7572 | 6.8 |
| ICD-10—Injury, poisoning and certain other consequences of external causes (yes) | 7559 | 6.8 |
| ICD-10—Congenital malformations, deformations and chromosomal abnormalities (yes) | 625 | 0.6 |
| | Median | IQR |
| Age (in years) | 70 | 24 |

CDS, Care Dependency Scale; ICD-10, International Statistical Classification of Diseases, 10th Revision.

patient-related fall risk factors (red line) or (b) patient-related fall risk factors as well as a non-linear time trend (green line). The red line shows that patient-related fall risk factors have tended to increase over time, which is reflected in increasing predicted risk-adjusted national fall rates. These are estimated to be 3.65% in the year 2011 and 4.04% in the year 2019. The increase in patient-related fall risk factors over time is also evident from online supplemental table S1. For 8 of the 10 variables included in the risk adjustment model, it was shown descriptively that the patient-related fall risk variables increased over the years, reaching their highest values in 2019, and concurrently the lowest values for the female sex and the percentage of completely care-independent patients in 2019. For example, the ICD-10 diagnosis group 'mental, behavioural and neurodevelopmental disorders' increased from 15.6% in 2011 to 20.5% in 2019. This underscores the necessity of risk adjustment. The green line shows a decrease followed by a slight increase in predicted national inpatient fall rates over time when changes in patient-related fall risk factors as well as a non-linear time trend are adjusted for. The predicted fall rates

based on this model are highest at 4.44% in 2011, lowest at 3.43% in 2016 and 3.73% in 2019. As none of the 95% CIs of the calculated ORs (online supplemental table S2) is above 1, the slight increase observed visually is not statistically significant.

In the additional analysis, a subsample consisting of 67 336 patients aged 65 years or older from 221 hospitals was considered. The results based on this subsample do not differ significantly from the overall sample: Although the national fall rates are descriptively consistently higher than in the overall sample, the same patient-related fall risk variables were selected into the risk adjustment model and the coefficients of the risk variables in the model were not substantially different (see online supplemental tables S3, S4 and figures S2,S3).

## Sensitivity analysis

The sensitivity analysis included 93 hospitals that provided complete data over all 9 survey years, with a total of 64 038 patients. In contrast to the overall sample, the results based on this subsample show a significant linear decreasing trend in the inpatient fall rates over time, both

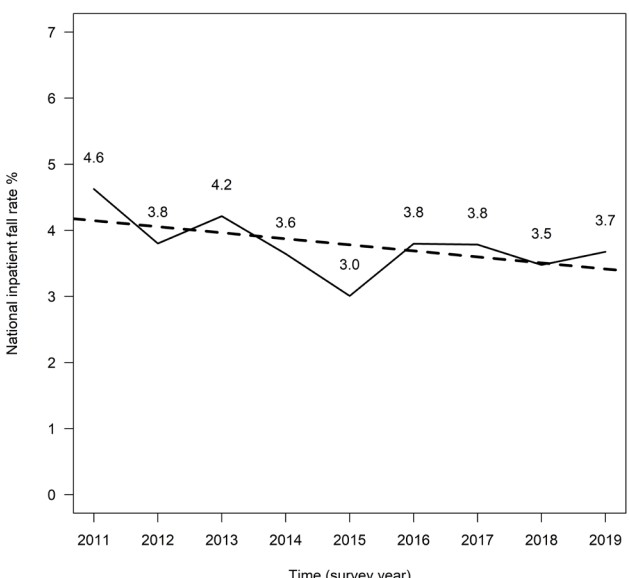

**Figure 1** Visualisation of the descriptive Swiss national inpatient fall rates from 2011 to 2019 and the estimated linear trend over time.

descriptively and risk-adjusted (online supplemental table S5, figures S4 and S5). Although the quadratic time effect was included in the model during AIC selection, the effect was not significant and was therefore omitted from the final model. In the risk adjustment model, with the exception of the additionally selected variable 'ICD-10—diseases of the circulatory system', the same patient-related fall risk variables were included in the model, and the coefficients of the risk variables in the model did not differ significantly from the model based on the overall sample (online supplemental table S5).

## DISCUSSION

In this study, we investigated whether inpatient fall rates in acute care hospitals in Switzerland have changed at the national level between 2011 and 2019, and whether these changes persist when patient-related fall risk factors are adjusted for. In the unadjusted analysis, a linear decreasing trend in inpatient fall rates at the national level was found across the years surveyed. In contrast, the risk-adjusted analysis (which controlled for the average effect of patient-related factors) revealed a significant non-linear decreasing trend across the same years. In other words, the effect appears to flatten out over time. Similar results, that is, a non-adjusted decreasing trend in fall rates, were found in studies based on NDNQI data in US hospitals over a 27-month period as well as 6-year time period.[40 41] More recently, another study found that combined inpatient falls and pressure injury rates in the USA decreased statistically significantly from 2010 to 2019 after risk adjustment.[8]

It is assumed that the negative trend in risk-adjusted national fall rates observed in our study between 2011 and 2019 is

largely due to continuous quality improvement of structures and processes for fall prevention in hospitals. This assumption is reinforced by a study which showed that improvements designed to prevent falls were implemented at both the structural and process levels in Swiss hospitals during the same time period.[21] For example, the average relative frequency of the number of fall prevention measures applied per patient, such as the evaluation of current medication or the evaluation of aid devices, increased significantly from 12.9% in 2011 to 23.1% in 2019.[21] The impetus for the improvement measures put in place by hospitals is likely to have been facilitated by the well-known quality improvement mechanisms associated with the introduction of national inpatient falls measurement in 2011. Although regular quality measurement alone does not automatically lead to quality improvement,[5] accompanying measures, such as public reporting,[6 42] comparative benchmarking[7] and performance feedback via quality dashboard,[43 44] are described as effective in triggering quality improvement activities in hospitals. The results of the present sensitivity analysis could also fit into this picture, in that hospitals that provided data on all nine survey years were more likely to have succeeded in continuously reducing their fall rates over time compared with the overall sample. It is possible that the regularity of quality measurement has led to a steady impetus to implement internal quality improvement measures. At least anecdotally, we are aware of some internal hospital projects aimed at reducing falls, which were initiated based on the results of the national quality measurement. Although it cannot be ruled out that other large-scale initiatives or programmes besides the introduction of the national quality measurement may have contributed to the reduction of falls in hospitals in Switzerland between 2011 and 2019, we are only aware of two fall prevention programmes of national scope implemented either before 2011 or after 2019 led by national organisations.[45 46]

Nevertheless, despite the quality improvement measures implemented by hospitals to prevent falls, the risk-adjusted national fall rates did not, as a whole, decline linearly over time. Instead, they showed a greater decline at the beginning of the surveys and then gradually levelled off, as demonstrated by the significant non-linear effect observed in our study. This pattern was also observed for nosocomial pressure injuries, another quality indicator of patient safety, at least descriptively over time.[16 21 47 48] For example, a decreasing trend that levelled off over time was observed in hospitals in the USA between 2006 and 2019[47] and in Switzerland between 2011 and 2019.[21 48] Therefore, the trend found in our study may not be specific to falls but may reflect a more general pattern. In principle, however, we can only speculate about the reasons for the non-linear trend found. Generally, it can be assumed that the lower fall rates get, the more difficult it becomes to reduce them further, as not all fall events are preventable.[13] Another explanation could be that, given the increase in patient-related fall risk factors over time, more and more effort is required on the part of hospitals just to keep fall rates constant and, accordingly, it becomes increasingly challenging to reduce them further.[21] This possible link is particularly emphasised by the results of the present

**Table 3** Overview of the two-level random intercept logistic regression model used to derive the risk-adjusted trend in Swiss national inpatient fall rates

| Predictors | Risk-adjusted trend in national inpatient fall rates | | | | |
| --- | --- | --- | --- | --- | --- |
| | Coeff. | SE | P value | OR | 95% CI |
| (Intercept) | −5.00 | 0.12 | **<0.001** | – | – |
| **Time-related factors (trend)** | | | | | |
| Time | −0.10 | 0.03 | **0.001** | – | – |
| (Time)$^2$ | 0.01 | 0.00 | **0.044** | – | – |
| **Patient-related fall risk factors** | | | | | |
| Age (in years) | 0.02 | 0.00 | **<0.001** | 1.02 | 1.01 to 1.02 |
| CDS (care independent (70–75)) | Ref. | | | | |
| CDS (to a great extent independent (60–69)) | 1.04 | 0.05 | **<0.001** | 2.83 | 2.57 to 3.11 |
| CDS (partially dependent (45–59)) | 1.42 | 0.05 | **<0.001** | 4.15 | 3.76 to 4.58 |
| CDS (to a great extent dependent (25–44)) | 1.74 | 0.06 | **<0.001** | 5.69 | 5.10 to 6.36 |
| CDS (completely dependent (15–24)) | 1.32 | 0.09 | **<0.001** | 3.73 | 3.15 to 4.42 |
| ICD-10—Mental and Behavioural disorders (yes) | 0.58 | 0.04 | **<0.001** | 1.79 | 1.66 to 1.92 |
| ICD-10—Neoplasms (yes) | 0.37 | 0.04 | **<0.001** | 1.45 | 1.35 to 1.56 |
| ICD-10—Diseases of the nervous system (yes) | 0.28 | 0.04 | **<0.001** | 1.32 | 1.22 to 1.44 |
| ICD-10—Diseases of the blood and blood-forming organs (yes) | 0.21 | 0.04 | **<0.001** | 1.23 | 1.13 to 1.33 |
| ICD-10—Injury, poisoning, other consequences of external causes (yes) | 0.15 | 0.06 | **0.007** | 1.16 | 1.04 to 1.30 |
| ICD-10—Endocrine, nutritional and metabolic diseases (yes) | 0.13 | 0.03 | **<0.001** | 1.13 | 1.06 to 1.21 |
| Sex (female) | −0.20 | 0.03 | **<0.001** | 0.82 | 0.77 to 0.87 |
| Surgical procedure within 14 days prior to measurement (yes) | −0.49 | 0.04 | **<0.001** | 0.61 | 0.57 to 0.66 |
| **Random effects** | | | | | |
| $\tau_{00}$ (variability in hospital intercepts) | | | 0.08 | | |
| N (hospitals) | | | 222 | | |
| Observations | | | 110 892 | | |

Significant p values are highlighted in bold.
CDS, Care Dependency Scale; ICD-10, International Statistical Classification of Diseases, 10th Revision.

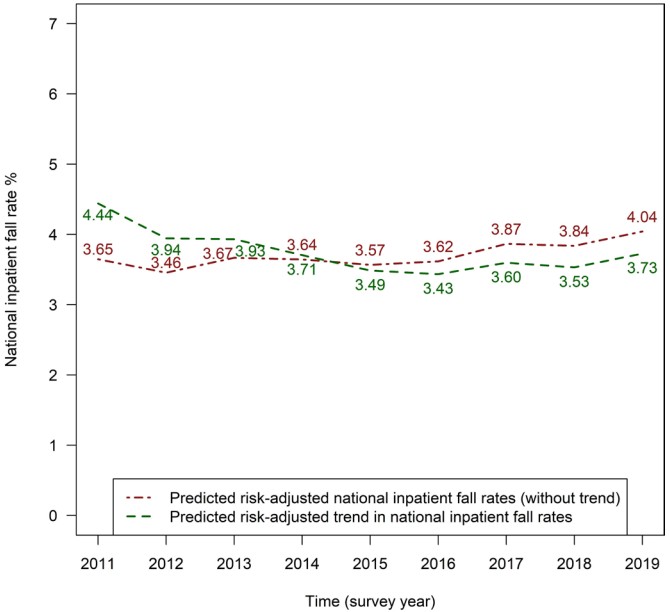

**Figure 2** Visualisation of predicted risk-adjusted (with and without time trend) Swiss national inpatient fall rates from 2011 to 2019.

study, which showed an increase in patient-related fall risk factors over time based on increasing predicted fall rates at the national level when prediction is based on these patient-related fall risk factors (as shown by the red line in figure 2). The observation that medical progress may unintentionally be contributing to a rise in risk factors and a concomitant levelling off of positive trends by enabling certain multi-morbid patients to survive has already been made with regard to pressure injuries.[21 47]

Almost more important than identifying the reasons for the non-linear trend is the fact that it indicates a potential increase in fall rates at the national level in the coming years if the observed trend continues. This development could be exacerbated by a further increase in fall risk factors and patient complexity in the hospital, as well as by a shortage of nursing staff, especially registered nurses, which is expected to worsen in the coming years both internationally[22] and in Switzerland.[49] A shortage of registered nurses has already been linked in a meta-analysis to poorer overall patient outcomes[50] and, in an observational study, to higher fall rates in the hospital.[51] Under these circumstances, it is strongly recommended that inpatient fall rates should be

continuously monitored. Such monitoring, on the one hand, would inform decision-makers at all levels and health professionals about potential deteriorations in the quality of care provided and enable initiation of appropriate data-driven countermeasures; on the other, it would determine whether the non-linear effect found here is confirmed in the future or whether it represents a sample-specific observation.

It should be noted that due to the large sample in our study, even small differences become statistically significant. Therefore, as in the study by Bouldin et al,[40] the question arises whether the observed decline in fall rates over time is also value based. In our study, we found an absolute reduction in predicted risk-adjusted fall rates of 0.71% between 2011 and 2019. Extrapolated to the slightly more than 1.2 million cases in general hospitals in 2019 in Switzerland,[52] this means that the quality improvement measures implemented over the years prevented 8520 fall events in 2019 compared with 2011. This decrease may seem small, but considering the possible impact of a fall on the individual, such as injury or death,[53] this decrease is nevertheless remarkable and of great value to patients. The decrease is also valuable from an economic point of view. If we multiply the prevented fall events by 4864 purchasing power parity adjusted international dollars, the average cost of a fall calculated by Morello et al,[12] it is estimated that more than 41 million international dollars in healthcare costs could have been saved nationwide in 2019. These savings underscore the economic benefits of investing in fall prevention strategies, for which a good return on investment is assumed.[10] Therefore, the implementation of evidence-based interventions, such as patient and staff education, could help to reduce fall rates or at least keep them constant.[54] This would not only have a positive impact on the direct cost savings already mentioned but also on the possibility of reallocating resources to other important areas of the healthcare system.

### Strength and limitations

A strength of our study is that the data used were collected using a highly standardised, internationally proven method and based on a census survey of all Swiss hospitals; it, thus, accurately reflects the situation at the national level. However, due to the average case inclusion rate of around 71% and the comparatively low rate of 36.1% in 2011, a possible selection bias in the LPZ measurement and in our study cannot be ruled out. A recall bias cannot be ruled out either, since the patients were asked whether they had fallen in the hospital in the last 30 days. As with the selection bias, the recall bias could have led to a potential underestimation of the fall rates observed. Since it can be assumed that the underestimation, if present, has remained constant over time, that is, a systematic underestimation, this would have no influence on the trend found. In general, it should be noted that adjustments can only be made for measurable risk factors for which data are available. Due to the limited number of patient-related fall risk factors collected with the LPZ questionnaire in Switzerland, not all potentially relevant risk factors could be considered in the risk adjustment model. For instance, it should be investigated whether it is possible to reliably collect

further data, for example, on frailty,[55] sarcopenia,[56] malnutrition,[57] impaired mobility,[58] urinary incontinence,[59] polypharmacy[60] or use of cardiovascular,[61] psychotropic,[62] opioid,[60] antiepileptic[60] medication at the national level without great effort. The effects of integrating some of these additional risk factors on the risk adjustment model would also need to be investigated. In addition, due to adjustments in the LPZ questionnaire over time, two relevant fall risk variables, a fall in the last 12 months or the use of sedatives/psychotropic medication (only collected from 2016 onwards) could not be taken into account in the risk adjustment model. Although these variables were not available for the present analyses, on the whole, the same patient-related characteristics with similar coefficients were identified as fall risk variables as in our previous studies based on LPZ data.[18 63] Due to the cross-sectional survey design, establishing a temporal link between the fall risk factors and the outcome is not possible. To enable this link and reduce selection bias, it is recommended that future trend analyses should look for opportunities to draw on data from the entire patient population, where possible. Ideally, these data should cover the entire period from admission to discharge and, where possible, the variables used for risk adjustment should relate to patient-related fall risk factors that were already present on admission. In this respect, electronic medical record data could in future provide a suitable data basis for national monitoring of inpatient falls over time, including risk adjustment.[64] However, further research is required.

### Conclusions

In our study, we found, descriptively, a significant linear decreasing trend in fall rates in Swiss hospitals and, after adjustment for patient-related fall risk factors, a significant non-linear decreasing trend, that is, a decrease that flattens out over time. It can be assumed that the introduction of the annual national measurement of inpatient falls in 2011 triggered improvement measures in hospitals in Switzerland, which in turn manifested in lower fall rates over time. The quality improvement achieved has had positive effects both for patients and for society as a whole by improving health and saving costs. However, the identified non-linear effect also showed that the fall rates decreased more at the beginning of the measurements than later on. This could be an indication that fall rates may increase in the future, especially in light of future challenges such as increasing fall risk factors, as shown in our study, and the staff shortages that hospitals are facing. The extent to which the observed non-linear trend will be confirmed in the future requires further investigation. In this context, it is recommended that national policy-makers continue to collect and monitor the fall rates in hospitals, on the one hand because this enables data-based quality development in the hospitals, and on the other so that a possible increase in national fall rates can be recognised at an early stage and appropriate targeted quality improvement measures can be planned and implemented at the national level. For this, robust, reliable and fair comparisons over time are important. Risk adjustment can be recommended as a method for improving the comparability of results over

time. To address the limitations of the present study, it is also recommended to further investigate to what extent routine clinical data are suitable as a basis for national comparisons of fall rates, including risk adjustment for patient-related fall risk factors to ensure comparability over time.

**Acknowledgements** We would like to thank all hospitals and patients who participated in the surveys.

**Contributors** Conceptualisation of the measurement: NSB, IHJE, SH and JMGAS. Conceptualisation of the manuscript: NSB and MM. Data collection and curation: NSB and MM. Data analysis: NSB and MM. Data interpretation: NSB, IHJE, SH, MM and JMGAS. Writing–original draft preparation: NSB; writing–review and editing, IHJE, SH, MM and JMGAS. NSB is responsible for the overall content of the manuscript as a guarantor.

**Funding** The annual national measurement of inpatient falls in Switzerland and thus the data collection was financed by the Swiss National Association for Quality Development in Hospitals and Clinics (ANQ) and carried out under mandate by the Bern University of Applied Sciences (BFH). The present data analysis was also financed by ANQ (Contract no. 'Kompensationsleistungen 2020').

**Competing interests** None declared.

**Patient and public involvement** Patients and/or the public were not involved in the design, or conduct, or reporting, or dissemination plans of this research.

**Patient consent for publication** Not applicable.

**Ethics approval** This study was conducted in accordance with the ethical principles of the Declaration of Helsinki. Full Research Ethics Committee approval for the primary data collection was granted by the lead Cantonal Ethics Committee Bern (Kantonale Ethikkommission Bern) on 4 October 2011 (application no. 122/11) in agreement with the other twelve local cantonal ethics committees: Kantonale Ethikkommission Thurgau, Kantonale Ethikkommission Luzern, Kantonale Ethikkommission Aargau, Kantonale Ethikkommission Zürich, Ethikkommission beider Basel, Ethikkommission des Kantons St. Gallen, Ethikkommission Appenzell Ausserhoden (AR), Commission d'éthique pour la recherche clinique en ambulatoire dans le canton de Genève, Commission d'éthique de recherche du canton de Fribourg, Commission cantonale valaisanne d'éthique médicale, Commission cantonale d'éthique de la recherche sur l'être humain, Comitato etico cantonale Ticino. As of 2012, Swiss ethics (Swiss Association of Research Ethics Committees) and the cantonal ethics committees declared the measurement as a quality measurement, which does not require ethical approval because it is not subjected to the Swiss Human Research Act. In the course of the reclassification of the measurement as a quality measurement, the ethics committees also agreed that written patient consent is no longer required, and therefore, written patient information followed by oral consent from the patients or their legal representatives is sufficient. Therefore, all patients or their legal representatives gave at least their verbal informed consent to participate in the measurement (see Bernet, *et al* (18) for more details). Furthermore, in 2022, the Ethics Advisory Board of the Bern University of Applied Sciences declared that the anonymised data from the LPZ measurement could be used again without further approval from a cantonal ethics committee (Ref. EAB2022_012). Only fully anonymised data were used in the present study.

**Provenance and peer review** Not commissioned; externally peer reviewed.

**Data availability statement** Data may be obtained from a third party and are not publicly available. The data on which the present results are based can be obtained on reasonable request from the Swiss National Association for Quality Development in Hospitals and Clinics (ANQ) (see www.anq.ch). However, the availability of these data may be subject to restrictions in accordance with the 'Empfehlungen Verwendung von ANQ-Daten zu Forschungszwecken (Recommendations for the use of ANQ data for research purposes)', which can be found at www.anq.ch.

**ORCID iD**
Niklaus S Bernet http://orcid.org/0000-0001-6478-1326

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
