## [Reviewer comments · BMJ Open]

ARTICLE DETAILS

TITLE (PROVISIONAL)	Risk-adjusted trend in national inpatient fall rates observed from 2011 to 2019 in acute care hospitals in Switzerland: a repeated multicentre cross-sectional study
AUTHORS	Bernet, Niklaus; Everink, Irma; Hahn, Sabine; Müller, Marianne; Schols, Jos

VERSION 1 – REVIEW

REVIEWER	Teles, Paulo João Figueiredo Cabral University of Porto
REVIEW RETURNED	15-Jan-2024

GENERAL COMMENTS	1. P12, line 247, “the hospitals as a random effect” – A random effect assumes random sampling from a larger population of levels (such as clusters like hospitals). However, the authors mention the study is based on “almost all Swiss hospitals with an acute somatic care mandate” (p. 3, line 63). If the data include almost all Swiss hospitals, it can hardly be considered a random effect since the sample almost coincides with the entire population. Or is the sample smaller than it appears? The text (lines 62-64) suggests the sample the sample is very close to the population. Perhaps the authors should make this point clearer and be more detailed about sample collection. Then, considering hospitals as a random effect would appear more appropriate and would raise no questions. 2. P. 15 Lines 299-305 – “decreasing trend” is not a U-shaped trend. Furthermore, a U-shaped curve contradicts “decreasing trend found flattens out over time”. In fact, a U-shaped curve starts by decreasing, reaches a minimum and then increases. This is not a decreasing trend and can distort the estimates of the fitted regression model. If a decreasing trend is required, other functions should be used such as $1/t$ or $\exp(-k/t)$, where k is a constant that can be estimated or set to provide a good fit. If, on the contrary, the authors really want a U-shaped trend, they have to make it clear in the text, rather than mentioning a “decreasing trend” that can be misleading. 3. P. 16 Table 3 – The authors mention they used stepwise backward selection of the predictors to be included in the regression model which is correct. But statistical significance of the estimated parameters should also determine the predictors to be kept in the model. Table 3 shows two nonsignificant risk factors: Certain infectious and parasitic diseases and Diseases of the genitourinary system. Therefore, the model should be fit again without these two
---

	predictors until all estimates are significant which will be the final estimated model. Lines 327-329 – The predicted fall rate (no trend) increases over time as shown in Figure 2 (red line) which means this model shows a poor fit and should be discarded, i.e., it is not an appropriate benchmark. The green line does show a better fit since the predicted rates follow the pattern of the actual ones. Furthermore, a U-shaped trend appears appropriate for the data and not a “decreasing trend” as is previously mentioned several times. A “decreasing trend” can be misleading as shown by the pattern of the fall rates since they decline in the beginning of the time period, level off and then start increasing in the last part of the period (U-shape). So, even though the rates in the latest years are still lower than those in the first years, a decreasing trend can be a risky claim because the data show a rebound. The authors should be more careful and mention such a pattern in the text (actually, it is briefly mentioned in the Discussion section in lines 388-390 and in the conclusions). If more recent data were available, fall rates could be close to the level of 2011 or 2013. A word of caution in the text would be appropriate. Another small remark: the range of the y axis in Figures 1 and 2 is too wide and makes changes appear very small. In Figure 1, a range from 2.5 to 5, for instance, would be enough. In figure 2, perhaps 3 to 5 would be appropriate. Then, figures do not need to be so large and take the whole page. The same applies to the supplementary figures. 4. A more general comment concerns patients’ age. Falls in hospital affect mostly older patients which means that this is a particularly serious problem for such patients. This study used all the data concerning falls regardless of age. Since the sample size is so large, I wonder whether this study could (and should) be restricted to older patients only (≥ 65 years, for instance) and the sample size would still be large. Perhaps results would be more significant and informative despite the lower sample size. From my point of view, I think the authors should have tried the same analysis with older patients only, even though the regression model considers age as a covariate and therefore adjusts for this variable. At least, I think it is worth giving it a try and compare the results.
--	---

REVIEWER	GE, LIXIA National Healthcare Group, Health Services and Outcomes Research
REVIEW RETURNED	16-Jan-2024

GENERAL COMMENTS	As a reviewer, I appreciate the thorough examination of inpatient fall rates in acute care hospitals in Switzerland between 2011 and 2019 and the efforts to understand the trends and potential contributing factors. Here are some comments and suggestions: Introduction In the introductory sentences, consider clarifying what is meant by "national quality measurements" and "change management". Specify whether they refer to specific metrics or indicators, respectively.
---

	Strengthen the connection between consecutive paragraphs for a smoother flow. For instance, you can link the idea of the positive impact of quality measurements to the subsequent discussion about falls in hospitals by highlighting the relevance of these measurements to adverse events. Where possible, add references to support your claims. For instance, when mentioning the positive influence of regular quality measurements combined with effective change management on the quality of care, you could cite specific studies. Line 98, it is good to indicate the currency of the cost \$6669. Ensure precision in the usage of terms. For example, in line 107, instead of "surveyed," you might use "monitored" or "tracked" for clarity. In line 115-117, where the authors mention the hypothesis that regular quality measurements favor change processes, consider briefly outlining the assumed link between quality measurements and change processes. This could enhance the reader's understanding of the underlying hypothesis. When discussing potential reasons for the stagnation of fall rates (lines 118-128), ensure clarity in the presentation of each reason. Consider breaking down the information into concise points or using bullet points to enhance readability. In a few sentences, consider streamlining the structure for conciseness. For instance, the authors might rephrase for clarity for the content in line 122-125. Line 130, good to describe what "this aspect" refers to since this is a new paragraph and there are many points described in the previous paragraph. Methods: There was a revision of the LPZ questionnaire during the data collection timeframe, which variables were affected and what is the impact of the revision? This should be described in detail. How the revision of the LPZ questionnaire affected the inpatient fall rates estimation should be clearly estimated. Although the authors have classified this study as a secondary data analysis, the detailed description of the data collection process provided in the Methods section suggests that the data were originally collected for primary purposes. Clarification on the primary objectives of the original data collection and how the current study utilizes these data for secondary analysis would enhance the transparency of the methodology. The authors mentioned "on the survey day" multiple times, why there is survey day since this is a secondary data analysis which should be conducted on pre-existing database? If the inpatient fall data were primarily collected via interviewing patients or their legal representatives, how did the hospitals manage to collect annual inpatient fall data simply on one day via survey? When was the data collected for an individual patient? What is the proportion of patients being surveyed annually? The detailed description of the LPZ measurement method is comprehensive (lines 150-164). However, consider breaking down complex sentences for readability. For example, the sentence starting with "To additionally ensure uniform data collection..." (line 165) could be divided into smaller sentences for clarity. The definition of an inpatient fall according to the LPZ measurement is provided (line 201). Consider moving this
--	--

	definition closer to the beginning of the Measures section for better contextualization. Clarify the transition from written consent in 2011 to verbal consent from 2012 onwards (lines 182-188). It might be beneficial to explicitly state the reason for this change in the consent process. Clearly outline the rationale behind selecting specific patient-related characteristics as potential variables for risk adjustment (lines 202). This could help justify the choice of variables and enhance the reader's understanding. Other than the listed variables, there are many other variables which may have better predictive power for inpatient falls, such as history of falls, history of stroke, medication use, polypharmacy, frailty status or mobility and gait issues, impaired vision, etc. The listed variables may not sufficiently capture the patient-level factors for inpatient falls. Was the study approved by any ethics review board? Data analysis Specify the criteria for cleaning and preparing the datasets uniformly. For instance, mention any exclusion criteria or methods used to handle missing data. Clarify the rationale behind using median and interquartile range (IQR) for age description (line 219). If there is a specific reason for choosing these measures over mean and standard deviation, it would be beneficial to mention it. Provide a brief explanation or reference for the method used to calculate annual fall rates (line 221). If there are specific considerations or adjustments made in this calculation, mention them. Before mentioning “the second research question” in Line 231, it will be good to state the specific research questions right after the purpose of the study. Lines 236-237, may I know what drove the authors to treat the survey year as a continuous variable instead of a categorical variable? Afterall, treating it a categorical variable will make the examination of the linear tend simpler. Offer more details about the stepwise backward variable selection algorithm based on the Akaike Information Criterion (AIC) (lines 242-244). Explain why this method was chosen and its implications for the study. Clarify the rationale for the sensitivity analysis focusing on hospitals that provided data in all nine survey years (line 263). Discuss any potential bias that might arise from hospitals joining or leaving the measurement during the study period. I am thinking whether it is appropriate to treat the selected variables as fixed effects in the model. Treating selected patient-level variables as fixed effects in a model is appropriate when you want to control for the variability associated with these variables and estimate their average effect across the entire population. However, it's crucial to consider the implications of this choice, especially when there are variations in these variables over the years. If the study's purpose is to account for the variability associated with these variables but assume that the coefficients follow a random distribution, including them as random effects should be used when you are interested in estimating the overall variance associated with these variables. If you suspect that the effects of patient-level variables may vary over the years, you might consider including interaction terms between these variables and the time variable. This allows you to assess whether the
--	---

	effects of these variables change significantly over the study period. Result For Table 2, since this study aimed to examine the trends of inpatient fall rates over the years, I will suggest the patient characteristics be presented by individual years instead of lumping together. This gives a better picture of the variation of patient characteristics over the years. This also allows for presenting participation rate for individual years in the table. Why the OR and 95% CI for time and time ² were not reported in Table 3? It will be good to describe how the risk adjusted fall rates were calculated in the Data analysis section. Figure 2 b) should the curvilinear time trend be adjusted when estimating the fall rates in individual years? Instead of saying “controlling for changing patient-related fall risk factors over time”, it is more appropriate to say “controlling for the average effect of patient-related factors over time” since they were included in the model as fixed effects. Discussion The identification of a curvilinear effect in the decline of fall rates is an intriguing finding. The discussion on the potential reasons for this trend is well-presented. However, it might be beneficial to explore potential contributing factors in more detail or discuss other studies that have observed similar trends. Line 385-386, the claim of an association between an increase in patient-related fall risk factors over time and the gradual leveling off of the decline in fall rates is an important observation. The authors should consider providing more information for this claim in the Result section and providing more insights into specific patient-related factors that contributed to this trend. The economic analysis is a strong point. It would be helpful to briefly discuss the economic implications of the findings in the context of the healthcare system, such as potential cost savings and resource allocation. The suggestion to include additional variables in future analyses, such as a temporal link between fall risk factors and outcomes, is valuable. Consider providing more specific recommendations for future research that can address the limitations identified in the study. The limited number of patient-level fall risk factors included in the models should be another limitation of the study. It would be beneficial if the authors could also indicate the changes in strategies applied in preventing inpatient falls or changes that might influence the inpatient fall rates over the years.
--	---

VERSION 1 – AUTHOR RESPONSE

Comments of Reviewer #1

1. Comment:

P12, line 247, “the hospitals as a random effect” – A random effect assumes random sampling from a larger population of levels (such as clusters like hospitals). However, the authors mention the study is based on “almost all Swiss hospitals with an acute somatic care mandate” (p. 3, line 63). If the data include almost all Swiss hospitals, it can hardly be considered a random effect since the sample almost coincides with the entire population. Or is the sample smaller than it appears? The text (lines 62-64) suggests the sample is very close to the population. Perhaps the authors should make this point clearer and be more detailed about sample collection. Then, considering hospitals as a random effect would appear more appropriate and would raise no questions.

Response:

*We view the model as the data generating mechanism, therefore the randomness does not come from sampling clusters or units. Because we are interested in estimating the model parameters (β coeff., τ_{00}), and not any finite population characteristics, a random effects approach is appropriate even if the sample contained the entire population (see e.g. S. Rabe-Hesketh and A. Skrondal. *Multilevel and Longitudinal Modeling Using Stata. Vol I, 4th edition, 2022, Stata Press*). However, in order to clarify and justify the decisionmaking process in the analytical procedure, we have adapted and supplemented the description in lines 334 to 337 as follows:*

“Although the sample comprised almost the entire population of hospitals, a random effects approach is recommended as this allows the estimation of model parameters instead of finite population characteristics.”³⁶

2. Comment:

P. 15, lines 299-305 – “decreasing trend” is not a U-shaped trend. Furthermore, a U-shaped curve contradicts “decreasing trend found flattens out over time”. In fact, a U-shaped curve starts by decreasing, reaches a minimum and then increases. This is not a decreasing trend and can distort the estimates of the fitted regression model. If a decreasing trend is required, other functions should be used such as $1/t$ or $\exp(-k/t)$, where k is a constant that can be estimated or set to provide a good fit. If, on the contrary, the authors really want a U-shaped trend, they have to make it clear in the text, rather than mentioning a “decreasing trend” that can be misleading.

Response:

Thank you very much for your valuable input. We completely agree with you. We have now completely omitted the term “U-shaped” in the manuscript. In addition, we have now consistently replaced the term “curvilinear” with “non-linear” in order to be more precise linguistically and avoid any misunderstandings. The adjustments led in particular to minor changes in the abstract, the data analysis section, the results section, the discussion and the conclusions. In the analysis section, in lines 319 to 324, we also included a justification for considering a quadratic term (which ultimately showed good fit) to determine a non-linear trend as follows: “and, to depict a trend, the time variable as a numerical covariate (coded from 1 to 9, where 1 denotes survey year 2011 and 9 survey year 2019). As it is rather unlikely that an increase or decrease will remain constant over the years, a quadratic time-effect was also included in model “A” in order to be able to take account of a non-linear relationship.”

3. Comment:

P. 16 Table 3 – The authors mention they used stepwise backward selection of the predictors to be included in the regression model which is correct. But statistical significance of the estimated parameters should also determine the predictors to be kept in the model. Table 3 shows two nonsignificant risk factors: Certain infectious and parasitic diseases and Diseases of the genitourinary system. Therefore, the model should be fit again without these two predictors until all estimates are significant which will be the final estimated model.

Lines 327-329 – The predicted fall rate (no trend) increases over time as shown in Figure 2 (red line) which means this model shows a poor fit and should be discarded, i.e., it is not an appropriate benchmark. The green line does show a better fit since the predicted rates follow the pattern of the actual ones. Furthermore, a U-shaped trend appears appropriate for the data and not a “decreasing trend” as is previously mentioned several times. A “decreasing trend” can be misleading as shown by the pattern of the fall rates since they decline in the beginning of the time period, level off and then start increasing in the last part of the period (U-shape). So, even though the rates in the latest years are still lower than those in the first years, a decreasing trend can be a risky claim because the data show a rebound. The authors should be more careful and mention such a pattern in the text (actually, it is briefly mentioned in the Discussion section in lines 388-390 and in the conclusions). If more recent data were available, fall rates could be close to the level of 2011 or 2013. A word of caution in the text would be appropriate.

Another small remark: the range of the y axis in Figures 1 and 2 is too wide and makes changes appear very small. In Figure 1, a range from 2.5 to 5, for instance, would be enough. In figure 2, perhaps 3 to 5 would be appropriate. Then, figures do not need to be so large and take the whole page. The same applies to the supplementary figures.

Response:

Thank you very much for the comments.

Regarding the comment on page 16, Table 3: *For the selection of variables, we decided in favour of the AIC criterion, as the mean square prediction error is smaller with this method compared to selection with the BIC criterion or “significant predictors only” and the AIC model is closer to the result with cross-validation. This comes at the price of more complex models. As a further reduction in model complexity is of course also justified, we have now examined whether the two non-significant variables can be omitted from the model in accordance with the reviewer’s comment. As the results have changed extremely marginally and the previous and the reduced model do not differ significantly in terms of model fit (ANOVA $p = 0.092$), we have now newly included the reduced model in the article. In the data analysis chapter, we have described the step described above in lines 338 to 345 as follows:*

“d) To determine whether the model complexity of model “C” could be further reduced, a further two-level random intercept logistic regression model (model “D”) was calculated. In model “D”, the hospitals were modelled as a random effect and the statistically significant time-related factors as well as the patient-related fall risk factors of model “C” were included as fixed effects. If the ANOVA test revealed no significant difference in the model fit between model “C” and the reduced model “D”, model “D” served as the final risk adjustment model that informed the subsequent steps e to f (see table 3).” **Regarding the comment on lines 327-329:** *Thank you for pointing this out. Our aim with the model without time effect was not to compare the two models with each other. As you rightly say, this model would also not be suitable as a reference model for assessing the fit of the model to data. With the model without time effect, we rather wanted to illustrate the development of patient characteristics over time. It also served as a basis for deciding whether risk adjustment for patient-related fall risk factors was necessary at all. The red line in the graph is purely hypothetical in this sense. However, we conclude from the commentary that the aim and purpose of the model without time effect is currently not sufficiently well described in the manuscript. We have therefore comprehensively revised the chapter on data analysis in lines 294 to 382 and tried to describe the individual steps more clearly. In particular, we have revised the section explaining the model without time effect in lines 346 to 356 as follows:*

“e) To determine whether the patient-related fall risk factors have changed over time and therefore to gain an idea of whether risk adjustment is necessary at all, the patient-related fall risk factors selected in model “D” (without time-related factors) were included as covariates in a two-level random intercept logistic regression model (model “E”). The need for risk adjustment is indicated if the patient-related fall risk factors have increased or decreased over time. This can be recognised if the national inpatient fall rates per survey year increase/decrease if they are predicted solely on the basis of the selected

patient-related fall risk factors. The risk-adjusted national inpatient fall rates predicted based on model “E” were graphically plotted over time for visualisation purposes (see red line in figure 2).”

In addition, we have tried to describe the results in the results section in lines 433 to 452 in analogy to the analysis steps. Here we have also tried to show more clearly that although an increase in fall rates is visually apparent towards the end, this is not statistically significant based on the 95% confidence intervals of the calculated odds ratios. Therefore, in our opinion, a decreasing non-linear trend in which the effect flattens out over time (and could possibly increase in the future) is the correct description. We have now described this consistently in this way in the discussion and in the conclusions.

Regarding the comment on the scaling of the y-axes in the figures: *We have adjusted the scaling of the y-axes in all figures.*

4. Comment:

A more general comment concerns patients’ age. Falls in hospital affect mostly older patients which means that this is a particularly serious problem for such patients. This study used all the data concerning falls regardless of age. Since the sample size is so large, I wonder whether this study could (and should) be restricted to older patients only (≥ 65 years, for instance) and the sample size would still be large. Perhaps results would be more significant and informative despite the lower sample size. From my point of view, I think the authors should have tried the same analysis with older patients only, even though the regression model considers age as a covariate and therefore adjusts for this variable. At least, I think it is worth giving it a try and compare the results.

Response:

Thank you very much for the tip. Following the comment, we have carried out the proposed extended analysis and added it to the manuscript as an additional analysis. In the analysis section, we have integrated the following paragraph in lines 366 to 368: “Fourth, an additional analysis was conducted by repeating the analyses on a subsample of patients aged 65 years and older. This subsample was selected because older patients in particular are more often subject to falls in hospital.”

The results of the additional analysis are presented in lines 453 to 459 as follows, and the corresponding tables and figures were added as supplementary table S3, table S4, figure S2 and figure S3:

“In the additional analysis, a subsample consisting of 67336 patients aged 65 years or older from 221 hospitals was considered. The results based on this subsample do not differ significantly from the overall sample: Although the national fall rates are descriptively consistently higher than in the overall sample, the same patient-related fall risk variables were selected into the risk adjustment model and the coefficients of the risk variables in the model were not substantially different (see supplementary table S3, table S4, figure S2 and figure S3).”

Comments of Reviewer #2

Introduction

1. Comment:

In the introductory sentences, consider clarifying what is meant by “national quality measurements” and “change management”. Specify whether they refer to specific metrics or indicators, respectively.

Response:

We have revised the first part of the introduction. We have tried to clarify the connection between national quality measurement and quality improvement using the framework of Berwick et al. 2003 in lines 80 to 100. In order to be more precise and avoid misunderstandings, we have now consistently referred to quality improvement processes instead of change management. After all, it is precisely

these processes that are to be initiated at the various levels of the healthcare system with the data collected through quality measurement. This comprehensive adjustment should also serve to clarify some of the subsequent remarks.

2. Comment:

Strengthen the connection between consecutive paragraphs for a smoother flow. For instance, you can link the idea of the positive impact of quality measurements to the subsequent discussion about falls in hospitals by highlighting the relevance of these measurements to adverse events.

Response:

Thank you very much for the comment. With the following sentence in line 101 to 102, for example, we have tried to link the two paragraphs more stringently as follows: "One of the quality indicators that could be improved through regular national measurements and targeted quality improvement efforts is inpatient fall rate." In addition, we had the entire revised manuscript checked again by a professional language proofreading service. It was explicitly pointed out that the links between the paragraphs should be checked to improve the reading flow and that suggestions for improvement should be added if necessary.

3. Comment:

Where possible, add references to support your claims. For instance, when mentioning the positive influence of regular quality measurements combined with effective change management on the quality of care, you could cite specific studies.

Response:

Thank you for your valuable feedback. We have carefully considered your suggestion. In revising our manuscript, we have restructured the introduction to address the issue you raised by clarifying the relationship between regular measurement, quality improvement strategies and their impact on quality of care. To ensure our claims are well-supported, we have included references to relevant systematic literature reviews that examine this relationship in detail. These references are now cited in lines 93 to 97 of our manuscript as follows:

"Regular quality measurement provides the basis for initiating data-driven quality improvement strategies such as benchmarking and public reporting or the monitoring of achievements over time.^{3 5} There is also some indication in the literature that these strategies can have a positive effect on the quality of care over time.^{6 7}"

4. Comment:

Line 98, it is good to indicate the currency of the cost \$6669.

Response:

Thank you for your valuable feedback. We have decided to be even more precise by converting the 6669 AUD into PPP-adjusted international dollars according to the 2022 OECD data, resulting in a cost of 4864 international dollars (6669:1.371=4864).

Therefore, we adapted the sentence in lines 104 to 107 as follows:

"Based on data from Australia, a fall in hospital results, on average, in an 8-day longer hospital stay and (converted into international dollars according to the Organisation for Economic Co-operation and Development¹¹) an additional cost of \$4864.¹²"

5. Comment:

Ensure precision in the usage of terms. For example, in line 107, instead of “surveyed”, you might use “monitored” or “tracked” for clarity.

Response:

Thank you very much for the comment. By analogy with the second comment above, we have had the entire manuscript checked by the professional proofreading service for precision in the use of the terms “surveyed” and “monitored” as well as “survey” and “measurement”.

6. Comment:

In line 115-117, where the authors mention the hypothesis that regular quality measurements favor change processes, consider briefly outlining the assumed link between quality measurements and change processes. This could enhance the reader's understanding of the underlying hypothesis.

Response:

Thank you for your suggestion. In response, we have introduced a new paragraph at the beginning of the introduction in lines 80 to 91, which explicitly outlines the hypothesized connection between national quality measurements and quality improvement. This addition aims to provide readers with a clear understanding of the foundational hypothesis that guides our study, highlighting how regular quality assessments can serve as a catalyst for initiating and sustaining quality improvement processes.

7. Comment:

When discussing potential reasons for the stagnation of fall rates (lines 118-128), ensure clarity in the presentation of each reason. Consider breaking down the information into concise points or using bullet points to enhance readability.

Response:

Thank you for your comment. We have more clearly separated the possible reasons for the stagnation of inpatient fall rates in lines 138 to 148 as recommended, and added bullet points as follows:

- *“Not all falls are preventable: Even if fall prevention is continuously improved, the fall rates will remain constant at a certain level, as a reduction to zero will not be achievable.¹³*
- *Lack of resources to intensify fall prevention: Due to increasing financial constraints and staff shortages, fall prevention cannot be further strengthened.²²*
- *Patient-related risk factors for falls are increasing: More and more often, older and multimorbid patients are being treated in the hospital, patients who carry a higher risk of falling. This means that more patients with a higher fall risk profile are being treated in hospital over time. As a result, falls in hospital do not decrease despite constant quality improvements.²¹”*

8. Comment:

In a few sentences, consider streamlining the structure for conciseness. For instance, the authors might rephrase for clarity for the content in line 122-125.

Response:

We have tried to implement the commentary (see previous comment). Additionally, we also asked the professional proofreading service to focus in particular on general sentence structure.

9. Comment:

Line 130, good to describe what “this aspect” refers to since this is a new paragraph and there are many points described in the previous paragraph.

Response:

Thanks for the tip. We have adjusted the sentence in lines 150 to 152 as follows: “In order to accurately compare inpatient fall rates over time while accounting for changing patient-related fall risk factors, it is usually recommended to conduct a riskadjusted comparison.”²³

Methods

1. Comment:

There was a revision of the LPZ questionnaire during the data collection timeframe, which variables were affected and what is the impact of the revision? This should be described in detail. How the revision of the LPZ questionnaire affected the inpatient fall rates estimation should be clearly estimated.

Response:

Thank you for the important feedback. We have described the effects of the revision of the LPZ questionnaire in 2016 in detail in lines 182 to 190 as follows:

“In 2016, it underwent a revision and was subsequently referred to as LPZ 2.0. This revision streamlined the questionnaire by reducing the number of structure and process indicators, while largely preserving the questions related to patient characteristics. Although the revision also affected the questions on quality indicators, these changes involved a reduction in the number of questions rather than a change in content. For instance, regarding the quality indicator on falls, specific context questions on the time and place of the fall and the main causes were deleted. Detailed descriptions of the adjustments made to the “inpatient fall” variable can be found in the paragraph “Outcome variable”.”

In addition, we have added more detailed information regarding the impact of the revision on the outcome variable in lines 235 to 272.

2. Comment:

Although the authors have classified this study as a secondary data analysis, the detailed description of the data collection process provided in the Methods section suggests that the data were originally collected for primary purposes. Clarification on the primary objectives of the original data collection and how the current study utilizes these data for secondary analysis would enhance the transparency of the methodology.

Response:

Thank you for your valuable comment requesting clarification on the nature of the data used in our study and the aims of its original collection. In order to make clearer which primary data our study is based on, and the original aim of its collection, we have tried to describe the data sources used more clearly in lines 114 to 125 in the introduction. We have also comprehensively revised the “Methods” section in order to make the distinction between primary data collection and secondary analysis clearer, for example by describing the criteria for case selection in detail.

3. Comment:

The authors mentioned “on the survey day” multiple times, why there is survey day since this is a secondary data analysis which should be conducted on pre-existing database? If the inpatient fall data were primarily collected via interviewing patients or their legal representatives, how did the hospitals

manage to collect annual inpatient fall data simply on one day via survey? When was the data collected for an individual patient? What is the proportion of patients being surveyed annually?

Response:

Thank you for your comment. In the comprehensive revision of the methods chapter (see also previous comment), we have emphasised more clearly the description of the original primary data collection using a questionnaire-based cross-sectional design, which was carried out on the second Tuesday of the second week in November among inpatients in Swiss hospitals. The fact that we have now also more clearly described the study population on which the secondary analysis is based should also contribute to clarification.

Comment:

4.

The detailed description of the LPZ measurement method is comprehensive (lines 150-164). However, consider breaking down complex sentences for readability. For example, the sentence starting with “To additionally ensure uniform data collection...” (line 165) could be divided into smaller sentences for clarity.

Response:

Thank you very much for the advice. The paragraph in lines 194 to 199 has been revised accordingly to improve readability.

5. Comment:

The definition of an inpatient fall according to the LPZ measurement is provided (line 201). Consider moving this definition closer to the beginning of the Measures section for better contextualization.

Response:

Many thanks for the tip. As part of the comprehensive revision of the methods chapter and the associated new “Outcome variable” chapter, the definition of a fall was moved to the beginning of this chapter (lines 236 to 238).

6. Comment:

Clarify the transition from written consent in 2011 to verbal consent from 2012 onwards (lines 182-188). It might be beneficial to explicitly state the reason for this change in the consent process.

Response:

Thank you very much for pointing this out. In line 209, we have included a corresponding reference to the chapter on Ethics statements. This describes the background that led to verbal informed consent being sufficient.

7. Comment:

Clearly outline the rationale behind selecting specific patient-related characteristics as potential variables for risk adjustment (lines 202). This could help justify the choice of variables and enhance the reader's understanding.

Response:

Thank you very much for the tip. In the new chapter “Patient-related fall risk factors (covariates)” in lines 273 to 286, we have now described in detail the process and the criteria used to select potential covariates for use in the risk adjustment model.

8.

Other than the listed variables, there are many other variables which may have better predictive power for inpatient falls, such as history of falls, history of stroke, medication use, polypharmacy, frailty status or mobility and gait issues, impaired vision, etc. The listed variables may not sufficiently capture the patient-level factors for inpatient falls.

Response:

We completely agree with your comment. We have expanded this aspect accordingly in the limitations in lines 581 to 594.

Comment:

9. Comment:

Was the study approved by any ethics review board?

Response:

Yes, this is described in the section Ethics statements with the subtitle Ethics approval in lines 652 to 683.

Data analysis

1. Comment:

Specify the criteria for cleaning and preparing the datasets uniformly. For instance, mention any exclusion criteria or methods used to handle missing data.

Response:

Thank you for your request to specify the criteria used for cleaning and preparing the datasets uniformly, including details on exclusion criteria and handling of missing data. In response to your comment, we outlined the steps taken in lines 295 to 298.

2. Comment:

Clarify the rationale behind using median and interquartile range (IQR) for age description (line 219). If there is a specific reason for choosing these measures over mean and standard deviation, it would be beneficial to mention it.

Response:

Thanks for the note. We have added the reason (skewed distribution) in lines 298 to 301.

3.

Provide a brief explanation or reference for the method used to calculate annual fall rates (line 221). If there are specific considerations or adjustments made in this calculation, mention them.

Response:

Thank you very much for the tip. We have clarified the paragraph in lines 301 to 306 accordingly and added a reference.

4. Comment:

Before mentioning “the second research question” in Line 231, it will be good to state the specific research questions right after the purpose of the study.

Response:

Thank you very much for the comment. We have decided not to formulate the objectives of the study as additional questions. Accordingly, we have reworded the sentence in lines 314 to 316 so that it does not include “to answer the second question” but rather refers to the second objective: “Third, to determine whether an identified trend persists after risk adjustment for patient-related fall risk factors, we performed the following steps: a to d (risk adjustment model development) and e to f (reporting and visualisation).”

Comment:

5. Comment:

Lines 236-237, may I know what drove the authors to treat the survey year as a continuous variable instead of a categorical variable? After all, treating it as a categorical variable will make the examination of the linear trend simpler.

Response:

Thank you for your comment. We chose to treat the survey year as a continuous variable to keep our model as simple as possible, based on the clear, albeit not perfectly linear, trend observed in the data. Starting with a linear term and then adding a quadratic term was sufficient to capture the trend effectively without unnecessary complexity. This approach aligns with our goal of model parsimony while still providing an accurate representation of the data trends.

6. Comment:

Offer more details about the stepwise backward variable selection algorithm based on the Akaike Information Criterion (AIC) (lines 242-244). Explain why this method was chosen and its implications for the study.

Comment:

Response:

Thank you for your comment. May we refer you to our response to comment 3 of Reviewer 1 on page 3 to 5. Here we have described in detail the reasons for the choice of the AIC as variable selection criterion. In addition, we have expanded the model selection in the article in accordance with Reviewer 1's comment, so we assume that the comment made here is likely to have been resolved.

7. Comment:

Clarify the rationale for the sensitivity analysis focusing on hospitals that provided data in all nine survey years (line 263). Discuss any potential bias that might arise from hospitals joining or leaving the measurement during the study period.

Response:

Thank you for pointing out the need for clarity in our sensitivity analysis. In the sensitivity analysis, we focussed on hospitals that provided data for all nine survey years to ensure consistency and reduce variability due to changing hospital cohorts. This stable sample enables an additional assessment of the stability of the trend found in the overall sample.

In the manuscript, we have added the following explanations in lines 372 to 377: "This approach was based on the assumption that the introduction of annual national quality measurement could trigger quality improvement initiatives in participating hospitals, potentially reducing fall rates over time. However, this trend could be masked by newer hospitals participating in the measurement, as they may initially have higher fall rates compared to hospitals already participating. The aim of the sensitivity analysis was therefore to detect such a masking effect, if one existed."

8. Comment:

I am thinking whether it is appropriate to treat the selected variables as fixed effects in the model. Treating selected patient-level variables as fixed effects in a model is appropriate when you want to control for the variability associated with these variables and estimate their average effect across the entire population. However, it's crucial to consider the implications of this choice, especially when there are variations in these variables over the years. If the study's purpose is to account for the variability associated with these variables but assume that the coefficients follow a random distribution, including them as random effects should be used when you are interested in estimating the overall variance associated with these variables. If you suspect that the effects of patient-level variables may vary over the years, you might consider including interaction terms between these variables and the time variable. This allows you to assess whether the effects of these variables change significantly over the study period.

Response:

Thank you for your thoughtful comment regarding our choice to treat selected patientlevel variables as fixed effects in our model. Our primary aim, as you correctly identified, was to control for the variability associated with these patient-level variables, and estimate their effects on the odds of a fall for an individual patient. This decision was driven by our specific interest in adjusting for these effects rather than estimating the overall variance that might be attributed to them.

However, you may also be referring to an average patient effect in the sense of whether the effect is constant across all hospitals and whether a random-intercept model is sufficient or whether it should be extended with random slopes. We have the impression that this extension (random slopes) is not necessary. In addition, we refrained from this extension due to the enormous computational time it would have taken. In the data analysis chapter, however, we have now consistently replaced the general term "multilevel logistic model" with the more specific term "two-level random intercept logistic regression model" in order to avoid any confusion.

Comment:

Thank you for your comments on testing an interaction effect between the patient-related fall risk variables and the time variable. Indeed, we used interaction terms between patient-related fall risk factors and time to examine variability over time. No significant interactions were found. This indicates a consistent impact of these variables on the odds of falling throughout the study. However, as this information is missing in the manuscript, we have added it accordingly in lines 324 to 325.

Results

1. Comment:

For Table 2, since this study aimed to examine the trends of inpatient fall rates over the years, I will suggest the patient characteristics be presented by individual years instead of lumping together. This gives a better picture of the variation of patient characteristics over the years. This also allows for presenting participation rate for individual years in the table.

Response:

Thank you for your valuable suggestion regarding the presentation of patient characteristics by individual years to better illustrate the variation over time. We have taken your advice into account and have now detailed patient characteristics across the survey years in supplementary table S1. Additionally, to maintain consistency with the newly inserted flowchart (supplementary figure S1), we have revised the terminology from “participation rate” to “case inclusion rate” within the text in lines 386 to 392. This change ensures coherence in our presentation of data and aligns with the methodology depicted in the flowchart, accurately reflecting the process of case selection and inclusion in our analysis.

2. Comment:

Why the OR and 95% CI for time and time ² were not reported in Table 3?

Response:

Thank you for your query regarding the absence of odds ratios (OR) and 95% confidence intervals (CI) for the time variable and its quadratic term (time²) in table 3. The calculation of ORs in the presence of both linear and quadratic terms for a variable is complex because the effect of a 1-unit increase in the time variable is not constant and depends on the value of the time variable itself due to the inclusion of the quadratic term. Therefore, separate ORs for the linear and quadratic components would not meaningfully represent the relationship between time and the outcome.

To accurately convey the combined effect of these terms, we decided to report only the regression parameters for time and time². We then calculated the overall OR for a 1-unit increase in time, considering both the linear and quadratic terms simultaneously, to provide a comprehensive view of how time affects the outcome variable. This approach ensures that the effect of time is correctly interpreted in the context of our model's specified relationship. For illustration of this nonlinear time effect, we refer readers to supplementary table S2, where we present the ORs for a 1-unit increase in time in a manner that reflects the nonlinear relationship between time and the inpatient fall rates.

3. Comment:

It will be good to describe how the risk adjusted fall rates were calculated in the Data analysis section.

Response:

Thank you for your suggestion to clarify the calculation of risk-adjusted fall rates in the data analysis section. Based on your and other valuable feedback, we have undertaken a thorough revision of this section in lines 314 to 365 to improve its comprehensiveness, clarity and structure. As a result of these extensive revisions, we believe that the approach to estimating risk-adjusted fall rates is now

Comment:

clear and concise. We believe that the revised section adequately addresses your concerns and therefore no further elaboration on this point is necessary. In addition, we have the impression that comment 5 below on page 17 could possibly help to clarify the uncertainties revealed by the present comment.

4. Comment:

Figure 2 b) should the curvilinear time trend be adjusted when estimating the fall rates in individual years?

Response:

Thank you for your inquiry regarding the adjustment of the curvilinear time trend when estimating fall rates in individual years, as mentioned for Figure 2b. We would like to clarify that our manuscript does not contain a Figure 2b, which may have led to some confusion. However, we could imagine that the question you've raised might relate to discussions addressed in our responses to comments 8 (page 14f), 3 (page 16) and 5 (page 17). We apologize for any confusion and are happy to provide further clarification if needed, based on the specific aspects of the analysis you're referring to.

5. Comment:

Instead of saying "controlling for changing patient-related fall risk factors over time", it is more appropriate to say "controlling for the average effect of patient-related factors over time" since they were included in the model as fixed effects.

Response:

Thank you for your comment regarding our wording for controlling patient-related fall risk factors over time. We have amended the sentence as you suggested, particularly to emphasise that we only used a random intercept model (see also comment 8 on page 14f).

Discussion

1. Comment:

The identification of a curvilinear effect in the decline of fall rates is an intriguing finding. The discussion on the potential reasons for this trend is well-presented. However, it might be beneficial to explore potential contributing factors in more detail or discuss other studies that have observed similar trends.

Response:

Thank you for highlighting the importance of further exploring the non-linear decreasing trend in fall rates and comparing it with trends observed in other studies. In response, we have expanded our discussion to include comparisons with similar patterns identified in nosocomial pressure injuries in lines 517 to 523 as follows to illustrate that this trend may not be unique to fall rates but could reflect a broader pattern in patient safety indicators: "This pattern was also observed for nosocomial pressure injuries, another quality indicator of patient safety, at least descriptively over time.^{16 21 51 52} For example, a decreasing trend that levelled off over time was observed in hospitals in the USA between 2006 and 2019⁵¹ and in Switzerland between 2011 and 2019.^{21 52} Therefore, the trend found in our study may not be specific to falls, but may reflect a more general pattern. In principle, however, we can only speculate about the reasons for the non-linear trend found."

Comment:

2. Comment:

Line 385-386, the claim of an association between an increase in patient-related fall risk factors over time and the gradual leveling off of the decline in fall rates is an important observation. The authors should consider providing more information for this claim in the Result section and providing more insights into specific patient-related factors that contributed to this trend.

Response:

Thank you for your insightful comment regarding the need for more detailed description on the association between the increase in patient-related fall risk factors over time and the observed trend in fall rates. In response, we have expanded our Results section in lines 438 to 446 to include a more comprehensive examination of how these risk factors have evolved as follows:

“The increase in patient-related fall risk factors over time is also evident from the supplementary table S1. For 8 of the 10 variables included in the risk adjustment model, it was shown descriptively that the patient-related fall risk variables have increased over the years, reaching the highest values in 2019 and the lowest values for female sex and percentage of completely care-independent patients in 2019. For example, the ICD-10 diagnosis group “Mental, behavioural and neurodevelopmental disorders” increased from 15.6% in 2011 to 20.5% in 2019. This underscores the necessity of risk adjustment.”

3. Comment:

The economic analysis is a strong point. It would be helpful to briefly discuss the economic implications of the findings in the context of the healthcare system, such as potential cost savings and resource allocation.

Response:

Thank you for recognizing the strength of the economic analysis in our study. We have expanded our discussion in lines 559 to 569 as follows to address the economic implications of our findings within the healthcare system context, highlighting the potential for significant cost savings and resource allocation:

“The decrease is also valuable from an economic point of view. If we multiply the prevented fall events by 4864 international dollars, the average cost of a fall calculated by Morello, et al. ¹², it is estimated that more than 41 million international dollars in healthcare costs could have been saved nationwide in 2019. These savings underscore the economic benefits of investing in fall prevention strategies, for which a good return on investment is assumed.¹⁰ Therefore, the implementation of evidence-based interventions, such as patient and staff education, could help to reduce fall rates, or at least keep them constant.⁵⁸ This would not only have a positive impact on the direct cost savings already mentioned, but also on the possibility of reallocating resources to other important areas of the healthcare system.”

Comment:

The suggestion to include additional variables in future analyses, such as a temporal link between fall risk factors and outcomes, is valuable. Consider providing more specific recommendations for future research that can address the limitations identified in the study.

Response:

Thank you for your comment. In lines 581 to 594 we have highlighted the problem of imperfect risk adjustment models and outlined possible additional risk factors that should be considered in the risk adjustment model. In addition, we described in lines 598 to 606 a possible approach to take account of the limitations of the present study with regard to selection bias and the lack of a temporal link between risk variables and outcome.

Comment:

4. Comment:

The limited number of patient-level fall risk factors included in the models should be another limitation of the study.

Response:

We have previously addressed this limitation in connection with Reviewer 2's comment 8 on the method (page 12) as well as the comment above.

5. Comment:

It would be beneficial if the authors could also indicate the changes in strategies applied in preventing inpatient falls or changes that might influence the inpatient fall rates over the years.

Response:

Thank you for the information. We have tried to address the comment accordingly in lines 501 to 512.

We believe that the revisions have significantly improved the quality of the manuscript. Attached, please find the revised version of the manuscript along with a marked-up copy highlighting the changes made. We hope that these modifications address the reviewers' concerns to your satisfaction.

Thank you once again for your time and valuable feedback. We look forward to hearing from you regarding the acceptance of our manuscript.

Sincerely,

The authors

VERSION 2 – REVIEW

REVIEWER	Teles, Paulo João Figueiredo Cabral University of Porto
REVIEW RETURNED	11-Mar-2024

GENERAL COMMENTS	The authors have successfully addressed my previous remarks and the paper is substantially improved. The two following minor comments are in order. 1. Line 336, "Although the sample...this allows the estimation of model parameters instead of finite population characteristics" – Any statistical model fitted to a sample, such as regression models, allows estimation of its parameters, regardless of being a random effects model or not. Using a sample is a source of randomness and any statistical model such as regression or ANOVA models include a disturbance term which is also a source of randomness. Therefore, this cannot be the justification for using a r.e. model. Furthermore, in this dataset, the units of analysis (patients) are nested within clusters (hospitals). If the clusters can be considered or have been sampled from a larger population of clusters, their effects can be modeled as random effects. This is what justifies a r.e. model. Anyway, not all Swiss hospitals were included in the
---

Comment:

	study, so they can be considered a r.e. or, even if they did, they could be considered a sample from the broader population of hospitals across the world. 2. Line 693, “international dollars” – do the authors mean USD? If they do, “international dollars” should be replaced by “US dollars” or “USD”.
--	---

REVIEWER	GE, LIXIA National Healthcare Group, Health Services and Outcomes Research
REVIEW RETURNED	14-Mar-2024

GENERAL COMMENTS	The authors have generally addressed the comments given earlier. Thanks for spending time and efforts addressing them. There are a few small points requiring further clarification. Introduction 1. I suggest the authors double checking the citations and references. The references 3 and 4 were cited for the US example initially, however, they were still cited for another different sentence/point although the US example was moved to the next paragraph, with another reference (reference 8) which was cited for something else. 2. “(converted into international dollar...Development) and additional cost of \$4864” can be changed to “and additional cost of \$4864 international dollars 11, to improve readability. Methods: 1. Page 12, Line 260-262 “Before admission (1) to 0...missing responses to 0”. Is it appropriate to straight away code missing responses as 0, which is used for before admission? Why not creating the new variable indicating whether the patient had fallen in the last 30 days by recoding the responses to “How often has the patient fallen in the last 30 days?” 2. Page 15. Line 311-313: “the Cochran-Armitage trend test was used...” The Cochran-Armitage trend is a statistical method used to analyze categorical data, particularly in the context of binary outcomes or proportions across ordered categories or groups. It assesses whether there is a linear trend or association between an ordinal independent variable (explanatory variable) and a binary dependent variable (outcome variable). If authors treated survey year as an ordinal variable, it is appropriate. However, the authors treated the survey year as a continuous variable instead of an ordinal variable, the Cochran-Armitage trend test sounds not appropriate. A logistic regression using year as a continuous variable for examining linear trend ($p < 0.05$) and a quadratic term ($p > 0.05$) might be sufficient. 3. Line 337 “a random effects approach” should be “a random-effects approach”.
---

VERSION 2 – AUTHOR RESPONSE

Comments of Reviewer #1

Comment:

1. Comment:

Line 336, “Although the sample...this allows the estimation of model parameters instead of finite population characteristics” – Any statistical model fitted to a sample, such as regression models, allows estimation of its parameters, regardless of being a random effects model or not. Using a sample is a source of randomness and any statistical model such as regression or ANOVA models include a disturbance term which is also a source of randomness. Therefore, this cannot be the justification for using a r.e. model. Furthermore, in this dataset, the units of analysis (patients) are nested within clusters (hospitals). If the clusters can be considered or have been sampled from a larger population of clusters, their effects can be modeled as random effects. This is what justifies a r.e. model. Anyway, not all Swiss hospitals were included in the study, so they can be considered a r.e. or, even if they did, they could be considered a sample from the broader population of hospitals across the world.

Response:

We thank you very much for your careful consideration of our reasons for choosing a random-effects model. We still believe that a random-effects model is appropriate, even if the sample contains (almost) the entire population because we are interested in estimating the random-effects model parameters (β coeff., τ_{00}), and not any finite population characteristics.

We must admit, however, that the sentence “Although the sample comprised almost the entire population of hospitals, a random effects approach is recommended as this allows the estimation of model parameters instead of finite population characteristics.[36]”, which was inserted in line 336f of the revised manuscript during last revision, is incomplete and can therefore be completely misunderstood. We would like to apologise for this. We propose to adjust the paragraph in lines 334 to 341 as follows in order to better explain the justification for the choice of a random-effects model: “c) Since the data have a hierarchical structure (patients grouped in hospitals), the selected variables according to model “B” were modelled as fixed effects and the hospitals as a random effect in a two-level random intercept logistic regression model (model “C”). We wanted to make inferences regarding the model parameters of a random-effects model, not any finite population characteristics. This model-based approach, in contrast to a designbased approach, allowed us to generalise to a hypothetical population beyond the data set under consideration.[36]”

2. Comment:

Line 693, “international dollars” – do the authors mean USD? If they do, “international dollars” should be replaced by “US dollars” or “USD”.

Response:

We have decided to convert the amount reported by the study in Australian dollars into purchasing power parity adjusted international dollars as a hypothetical reference currency. According to the World Bank website

<https://datahelpdesk.worldbank.org/knowledgebase/articles/114944-what-is-aninternational-dollar>, international dollars are defined as follows: “An international dollar would buy in the cited country a comparable amount of goods and services a U.S. dollar would buy in the United States. This term is

Comment:

often used in conjunction with Purchasing Power Parity (PPP) data". This made it easier for an international readership to interpret or convert the avoided costs reported in our study.

To emphasise more clearly that a transformed currency is used here, we have added "purchasing power parity adjusted international dollars" to the sentence in lines 567 to 571.

Comments of Reviewer #2

Introduction

1. Comment:

I suggest the authors double checking the citations and references. The references 3 and 4 were cited for the US example initially, however, they were still cited for another different sentence/point although the US example was moved to the next paragraph, with another reference (reference 8) which was cited for something else.

Response:

Thank you for pointing this out. We have double-checked the references and citations again.

In the initially submitted version, the sentence "For example, in the US, a risk-adjusted trend analysis showed a significant reduction in adverse events in hospitals between 2010 and 2019.[4]" included reference 4, which refers to the following source: 4. Eldridge N, Wang Y, Metersky M, et al. Trends in Adverse Event Rates in Hospitalised Patients,

2010-2019. Jama 2022;328(2):173-83. doi: <https://doi.org/10.1001/jama.2022.9600> In the revised version of the article, the numbers of the references have changed, as we have added a new paragraph above and the references in the text are numbered sequentially according to BMJ reference style (Vancouver style). In the revised manuscript, reference 8 was added to the sentence "For example, a risk-adjusted trend analysis in the US between 2010 and 2019 showed a significant reduction in adverse events, particularly in hospitals that were affected by targeted quality improvement efforts during this period.[8]", which also refers to the following source: 8. Eldridge N, Wang Y, Metersky M, et al. Trends in Adverse Event Rates in Hospitalised Patients, 2010-2019.

Jama 2022;328(2):173-83. doi: <https://doi.org/10.1001/jama.2022.9600>

As we could not identify any discrepancies, we have not made any adjustments.

2. Comment:

Comment:

“(converted into international dollar...Development) and additional cost of \$4864” can be changed to “and additional cost of \$4864 international dollars 11, to improve readability.

Response:

Thanks for the suggestion. As we have the impression that not all readers are familiar with international dollars as a hypothetical reference currency, we have added “purchasing power parity adjusted international dollars” in line 104f to make it clear that this is a transformed standardised currency (see also our response to comment 2 of Reviewer 1 on page 3 for more information).

Methods

1. Comment:

Page 12, Line 260-262 “Before admission (1) to 0...missing responses to 0”. Is it appropriate to straight away code missing responses as 0, which is used for before admission? Why not creating the new variable indicating whether the patient had fallen in the last 30 days by recoding the responses to “How often has the patient fallen in the last 30 days?”

Response:

Thank you for your comment. We agree with you that the previous procedure for creating the outcome variable in the years 2011 to 2015 was not ideal. We have now adapted the recoding based on your comment and described it accordingly in the text in lines 258 to 266. The review of the new procedure for recoding to form the outcome variable actually revealed a marginal difference to the previous procedure in that an additional case in the 2012 data set had to be excluded from the analysis sample due to missing information.

The numbers were adjusted accordingly at the relevant points.

2. Comment:

Page 15. Line 311-313: “the Cochran-Armitage trend test was used...” The Cochran-Armitage trend is a statistical method used to analyze categorical data, particularly in the context of binary outcomes or proportions across ordered categories or groups. It assesses whether there is a linear trend or association between an ordinal independent variable (explanatory variable) and a binary dependent variable (outcome variable). If authors treated survey year as an ordinal variable, it is appropriate. However, the authors treated the survey year as a continuous variable instead of an ordinal variable, the Cochran-Armitage trend test sounds not appropriate. A logistic regression using year as a continuous variable for examining linear trend ($p < 0.05$) and a quadratic term ($p > 0.05$) might be sufficient.

Comment:

Response:

We thank you very much for your comment. We chose the Cochran Armitage Trend test as the simplest method of analysis for the type of data and question we had as a starting point. As this test is still widely used, we also ran it for the sake of completeness. In this analysis, the variable “time” was treated as an ordinal variable. In order to describe these considerations and the handling of the time variable more precisely and in greater detail, we have adapted the paragraph in lines 308 to 314 along the commentary as follows: “Second, to investigate whether a trend in the prevalence of inpatient falls in acute care hospitals could be statistically confirmed at the national level in Switzerland between 2011 and 2019, we started the analysis with the simple and still frequently used Cochran-Armitage trend test. This test is suitable for identifying a linear trend or association between an ordinal dependent variable (time treated as an ordinal variable here) and a binary dependent variable, where the null hypothesis is that there is no trend,[34] i.e., that fall rates have remained the same over time.”

In the second step, we then moved on to the more complex methods with the logistic (multilevel) models, which allowed us to analyse the development of national fall rates over time in more detail. As explained in the last revision in the point-by-point protocol on page 13, we treated the variable “time” as a continuous variable in these analyses in order to keep the number of parameters to be estimated small (model parsimony). In addition, the overall aim was to look for a trend, whereby it was irrelevant whether this was linear or non-linear; accordingly, the identification of differences between two years, which would have made it necessary to include the variable “time” as an ordinal variable, was not a priority.

3. Comment:

Line 337 “a random effects approach” should be “a random-effects approach”.

Response:

Thanks for the suggestion. We have implemented this accordingly in line 338.

We believe that the revisions have significantly improved the quality of the manuscript. We hope that these modifications address the reviewers' concerns to your satisfaction.

Thank you once again for your time and valuable feedback. We look forward to hearing from you regarding the acceptance of our manuscript.

Sincerely,

Comment:

The authors

VERSION 3 – REVIEW

REVIEWER	Teles, Paulo João Figueiredo Cabral University of Porto
REVIEW RETURNED	25-Mar-2024
GENERAL COMMENTS	The authors have addressed my prior comments and remarks very well. The paper can be accepted for publication.